# MOF-Derived Co Nanoparticles Catalyst Assisted by F- and N-Doped Carbon Quantum Dots for Oxygen Reduction

**DOI:** 10.3390/nano13142093

**Published:** 2023-07-18

**Authors:** Yuqi Ma, Ki-Wook Sung, Hyo-Jin Ahn

**Affiliations:** Department of Materials Science and Engineering, Seoul National University of Science and Technology, Seoul 01811, Republic of Korea; yqma@seoultech.ac.kr (Y.M.); sung-kiuk@seoultech.ac.kr (K.-W.S.)

**Keywords:** carbon quantum dots, heteroatom co-doping, oxygen reduction reaction, metal–organic framework, oxygen reduction reaction catalytic activity

## Abstract

The oxygen reduction reaction is crucial in the cathode of fuel cells and metal–air batteries. Consequently, designing robust and durable ORR catalysts is vital to developing metal–air batteries and fuel cells. Metal–organic frameworks feature an adjustable structure, a periodic porosity, and a large specific surface area, endowing their derivative materials with a unique structure. In this study, F and N co-doped on the carbon support surface (Co/FN-C) via the pyrolysis of ZIF-67 as a sacrificial template while using Co/FN-C as the non-noble metal catalysts. The Co/FN-C displays excellent long-term durability and electrochemical catalytic performance in acidic solutions. These performance improvements are achieved because the CQDs alleviate the structural collapse during the pyrolysis of ZIF-67, which increases the active sites in the Co nanoparticles. Moreover, F- and N-doping improves the catalytic activity of the carbon support by providing additional electrons and active sites. Furthermore, F anions are redox-stable ligands that exhibit long-term operational stability. Therefore, the well-dispersed Co NPs on the surface of the Co/FN-C are promising as the non-noble metal catalysts for ORR.

## 1. Introduction

Owing to the limited reserves and environmental pollution worldwide, concerns regarding fossil fuel depletion have increased. Therefore, highly efficient, renewable, and clean energy sources have received increasing attention [1,2,3]. Among the various sources of regenerative energy, fuel cells and metal–air batteries are the best prospective chemical energy sources because of their low cost, sizeable energy-converting efficiency, and large theoretical energy density [4,5]. The oxygen reduction reaction (ORR) is one of the two half-reactions indispensable for energy transformation and storage. However, the ORR presents some issues, such as slow electron transfer, low mass transfer efficiency, and slow kinetics, which hinder the practical development of these devices. Therefore, sufficient, stable, and efficient catalysts for the ORR are essential for the efficient operation of the abovementioned devices [6,7]. Among the various catalysts, Pt-based catalysts have attracted widespread attention for their high selectivity and large electrocatalytic activity. High price and low stability hinder their application in energy storage devices [8,9]. Therefore, the non-noble metal catalyst has been extensively explored because of its low cost and excellent stability [10,11,12]. Despite these advantages, non-noble metal catalysts present critical issues, for example, low catalytic activity, i.e., lower than noble metal-based catalysts. Therefore, the practical design and fabrication of non-noble metal catalysts are crucial to enhancing the catalytic activity of ORRs.

Co-metal catalyst has been extensively studied among non-noble metal catalysts because of their benefits, such as abundant resources, low preparation cost, and bifunctional catalytic activity [13,14,15,16,17]. However, Co-metal catalysts present some limitations, such as low selectivity and stability, which impede their advanced application [18,19]. Li et al. directly grew Co_3_O_4_ with Fe-doped nanosheets on Ni foam to overcome these limitations. The synergistic action between Fe and Co atoms enabled an extremely low overpotential and high oxygen-evolving reaction (OER) stability [20]. Huang et al. introduced Mo atoms into a cobalt-based metal–organic framework (MOF) to construct a hierarchical microstructure of CoOx-MoC/N-doped carbon, which showed excellent long-term stability and outstanding OER catalytic activity [21]. Nevertheless, increasing stability and electrocatalytic activity by optimizing the morphology of the catalyst requires further investigation. As self-sacrificing templates, zeolitic imidazolate framework (ZIF), a type of MOF, has been applied to obtain Co-based electrocatalysts with carbon supports via pyrolysis [22]. Furthermore, Co-based electrocatalysts derived from ZIFs exhibit stable catalytic performance in acidic or alkaline electrolytes [23,24]. Especially, Co-NC catalysts involving Co-N_x_ species manifest significant ORR performances in acidic solutions due to the bonding of nitrogen atoms with cobalt atoms in their porphyrin-like structure. However, due to each atom’s high specific surface energy during pyrolysis, Co agglomerates significantly efficiently reduce the utilization of the metal active site [25]. Simultaneously, the polyhedral shape of the ZIFs collapses, which hinders the active site exposure and reduces electrocatalytic activity [26]. Therefore, to maintain the polyhedral structure of the ZIF during pyrolysis, it should be developed as an effective electrocatalyst for ORR. In this regard, well-dispersed electrocatalysts on carbon support that maintains a polyhedral structure and possesses high electrocatalytic activity and stability must be achieved.

Herein, we propose well-dispersed Co nanoparticles (NPs) on an F and N co-doping carbon support (Co/FN-C) by mixing F and N co-doping carbon quantum dots (FNCQDs) with ZIF-67 and performing pyrolysis for non-noble metal catalysts for ORR. Notably, the carbon quantum dots (CQDs) comprise abundant functional groups on their surfaces, which can protect the carbon structure and stabilize the metal NPs during pyrolysis. Therefore, the Co NPs are well distributed on the carbon support surface. Furthermore, the FNCQDs provide F and N atoms into the carbon support. The well-dispersed Co NPs increase electrocatalytic active sites, and F- and N-doping in the carbon support enhances ORR’s catalytic stability and activity.

## 2. Materials and Methods

FNCQDs were fabricated through a hydrothermal process. Specifically, urea (1.24 g, 99.5%, Sigma, St. Louis, MO, USA), sodium fluoride (0.9 g, 99.5%, Sigma), and citric acid (2.73 g, 99%, Sigma) were dissolved entirely in deionized water. The solvent was put into a hydrothermal synthesis autoclave lined with Teflon and heat-treated at 180 °C for 6 h. The obtained solvent was centrifuged thrice (30 min, 10,000 rpm) and then dialyzed for 24 h through a cut-off membrane (molecular weight of 6–8 kD). Finally, to fabricate FNCQDs, the dialysate was dried at 50 °C. The same procedure fabricated N-doped CQDs (NCQDs) without NaF to confirm the effect of F doping. To fabricate an F and N co-doped carbon support (Co/FN-C), 2-methylimidazole (99%, Sigma), cobalt (II) nitrate hexahydrate (98%, Sigma), and FNCQDs were stirred for 6 h in 23 mL of deionized water and obtained after three rounds of centrifugation (5 min, 5000 rpm). The precipitate was dried overnight at 50 °C. In addition, NCQD/ZIF-67 was obtained by replacing the FNCQDs with NCQDs using the same method. Finally, Co/C, Co/N-C, and Co/FN-C were prepared by heating ZIF-67, NCQD/ZIF-67, and FNCQD/ZIF-67 at 750 °C in an Ar atmosphere for 2 h.

The structure, morphology, and elemental distribution of the fabricated sample were investigated via transmission electron microscopy (TEM, Tecnai G2, FEI, Hillsboro, OR, USA), energy dispersive spectroscopy (EDS, Tecnai G2, FEI, Hillsboro, OR, USA) mapping, and field emission scanning electron microscopy (EVO10, Carl Zeiss, FESEM, Oberkochen, Germany), and the surface functional groups, chemical bonding states, and crystal structure of the sample was investigated via Raman spectroscopy (NRS-5100, JASCO, Tokyo, Japan), X-ray photoelectron spectroscopy (XPS, ESCALAB 250, Thermo Fisher, Waltham, MA, USA), and X-ray diffraction (XRD, Rigaku D/Max-2500 diffractometer, Rigaku, Tokyo, Japan), respectively. Specific surface area and pore structure determined of the sample was investigated by Brunauer–Emmett–Teller (BET, BELSORP-mini II, BEL Japan INC, Osaka, Japan).

Potentiostat/galvanostat (Ecochemie Autolab, PGST302N, Metrohm, Utrecht, The Netherlands) for measuring the electrochemical performance of catalysts. This potentiostat/galvanostat has a rotating disc electrode and an electrochemical workstation with a speed controller. The three-electrode assembly comprised a working electrode, a reference electrode, and a counter electrode. Homogeneous inks, which contained 10 mg of the fabricated samples as a catalyst in Nafion^®^ perfluorinated resin solution (57.2 μL), deionized water (50 μL), and 2-propanol (900 μL), were prepared via dispersion through sonication 30 min and string 12 h. The ink (1.8 µL) was dispersed on the working electrode surface (area: 0.0706 cm^2^) and dried (30 min, 50 °C) before the measurement was performed. Cyclic voltammetry (CV) was performed in Ar and O_2_ saturated 0.1 M HClO_4_ electrolyte. Linear sweep voltammetry (LSV) was measured at rotational speeds of 100, 400, 900, and 1600 and a scanning rate of 5 mV/s in O_2_-saturated 0.1 M HClO_4_ electrolyte. Finally, the accelerated durability test (ADT) was tested at a scanning rate of 100 mV/s for 5000 cycles to confirm the long-term electrocatalytic stability. LSV results were re-tested again at 1600 rpm after the ADT.

## 3. Result and Discussion

Figure 1 illustrates a diagram graphic of the fabricating method for the (a) FNCQDs, (b) FNCQD/ZIF-67, and (c) Co/FN-C. The FNCQDs were fabricated through a hydrothermal process. Urea provided the N source, sodium fluoride provided the F source, and citric acid provided the C source. FNCQD/ZIF-67 was fabricated by stirring FNCQDs and ZIF-67. Finally, F- and N-doped carbon-supported cobalt NPs were obtained via the pyrolysis of FNCQD/ZIF-67. During the pyrolysis, carbon support was formed from ZIF-67 as a sacrificial template, and Co NPs were forming on the surface of carbon support owing to the diffusion of Co atoms in ZIF-67. The CQDs effectively maintained the structure of the sacrificial template of ZIF-67; therefore, the Co NPs were well dispersed on the carbon support surface, thus increasing electrocatalytic active sites. Furthermore, F and N were doped into the carbon support via F and N atoms from the FNCQDs during pyrolysis, which enhanced the catalytic stability and ORR activity of the Co/FN-C.

Appendix A show the TEM and EDS mapping of NCQD and FNCQD. The NCQD and FNCQD particles were uniform in size with a diameter of ~5.0 nm (Appendix A) and the lattice fringe space is about 0.32 nm, representing the (002) plane of graphite (Appendix A). In Appendix A, the EDS mapping indicated that C and N atoms were present in the NCQDs. Meanwhile, as shown in Appendix A, the EDS results indicated that C, F, and N existed in the FNCQDs. This indicates that we have successfully prepared F- and N-doped CQDS.

To confirm the elemental distribution, the TEM-EDS mapping of the (a) ZIF-67, (b) NCQD/ZIF-67, and (c) FNCQD/ZIF-67 was performed (Appendix A). The TEM images of the three samples exhibited a polygonal structure with a size ranging from ~365 to ~395 nm. The EDS mapping indicated that C, Co, and N existed in the ZIF-67, NCQD/ZIF-67, and FNCQD/ZIF-67. Furthermore, only FNCQD/ZIF-67 showed F atoms. This shows that F atoms can be doped at Co/FN-C after pyrolysis of FNCQD/ZIF-67.

As shown in Appendix A, the XRD pattern was analyzed to check the crystal structures of ZIF-67, NCQD/ZIF-67, and FNCQD/ZIF-67. All samples exhibited sharp diffraction peaks at 7.4°, 10.4°, 12.6°, 14.7°, 16.5°, 18.1°, 22.2°, 24.5°, 25.6°, 26.7°, 29.6°, 31.5°, and 32.4° which was attributed to the (011), (002), (112), (022), (013), (222), (114), (233), (002), (134), (044), (244), and (235) planes of simulate ZIF-67 [27]. The intensity of the diffraction peak at 12.6° for NCQD/ZIF-67 and FNCQD/ZIF-67 nanocomposites is lower than that of pure ZIF-67, suggesting that the crystal structure of ZIF-67 changes with the formation of the composites. Appendix A shows the N_2_ adsorption–desorption isotherms of ZIF-67, NCQD/ZIF-67, and FNCQD/ZIF-67. Three samples all displayed type-I isotherms. The BET surface areas of ZIF-67, NCQD/ZIF-67, and FNCQD/ZIF-67 were 1152.2, 1166.6, and 1158.5 m^2^ g^−1^, respectively (Appendix A). The results showed that the surface areas of ZIF-67, NCQD/ZIF-67, and FNCQD/ZIF-67 were almost the same and had a higher specific surface area [28].

FESEM was conducted on all samples to examine their morphology (see Figure 2). Figure 3a shows that Co/C sizes of ~295 to ~353 nm exhibited a collapsed polyhedral structure. During the pyrolysis of ZIF-67, shrinkage occurred because of the dehydration of the organic components under anaerobic conditions [29,30]. Furthermore, the Co NPs agglomerated because of the significant surface energy of the single atoms, significantly reducing the active sites’ efficiency utilization [31]. However, Co/N-C and Co/FN-C (Figure 2a,b, respectively), exhibited polyhedral structures and bumpy surfaces with sizes of ~350 to ~410 nm. The unchanged polyhedral structure of Co/FN-C and Co/N-C was due to the CQDs, which served as protective layers. The plentiful oxidized functional groups of the CQDs strongly coordinate with the Co nodes in ZIF-67. Subsequently, ZIF-67 is pyrolyzed, and the CQDs interact strongly with ZIF-67 alleviating the structure collapse while preventing the severe aggregation of adjacent Co, thus affording well-dispersed Co NPs on the carbon support surface [32,33,34]. This confirmed that carbon support derived from ZIF-67 as a sacrificial template was fabricated successfully and enabled a polyhedral structure to be maintained via the CQDs.

To characterize the nanostructures, a TEM analysis of the Co/FN-C was performed. As shown in Figure 3a, relatively dark spots of Co NPs in the size range of ~15 to ~52 nm were observed with the relatively bright region of the carbon support, which exhibited a polygonal structure with sizes from ~378 to ~398 nm. Furthermore, Figure 3b shows that the bright region featuring the lattice fringe space is about 0.34 nm, representing the (002) plane of graphite, whereas the dark spot featuring the lattice fringe space is about 0.17 nm, representing the (200) plane of Co. To confirm the elemental distribution, a TEM-EDS mapping of the Co/FN-C was performed (Figure 3c). The EDS results indicated that C, N, F, and Co were uniform dispersion along the NPs and carbon support. These results prove that the Co NPs were well dispersed in the Co/FN-C, thus enhancing the electrocatalytic activity by increasing the active sites in the Co NPs.

Figure 4a shows that the XRD pattern was analyzed to investigate the crystal structures of three samples. All samples showed a broad peak of diffraction at ~24.6°, which corresponds to the (002) plane of graphite (JCPDS 65-6212), and sharp peaks of diffraction at ~75.9°, ~51.3°, and ~44.2°, which corresponds to the (220), (200), and (111) planes of Co (JCPDS 15-0806) [35,36]. Peaks other than those of amorphous graphite and metallic Co were not observed, indicating that all the Co^2+^ in ZIF-67 was completely reduced to metallic Co during pyrolysis. The graphitization degrees of three samples were determined using Raman spectroscopy (Figure 4b). Three samples exhibited two characteristic Raman absorption peaks within the compass of 1000–1800 cm^−1^ (1350 cm^−1^: D-band absorption peak, 1580 cm^−1^: G-band absorption peak). Theoretically, the G bands belong to the order of the degree of the sp^2^ carbon graphitic structures, and the D bands represent the disorder and defects of the sp^3^ hybridized carbon. The material’s graphitizing degree is compared through I_D_/I_G_ [37,38]. Compared with Co/C (1.010), Co/FN-C and Co/N-C exhibited significantly large I_D_/I_G_ of 1.032 and 1.028, suggesting that the F and N atoms doping induced defects in the carbon lattice and exposed the edge plane. ORR activity was generated by activating the π electrons, which was achieved by disrupting the integrity of its π-conjugated structure. Therefore, F and N doping in sp^2^ carbons resulted in defects by breaking the integrity of its π-conjugated structure, thus enhancing the activity of the ORR [39,40,41,42].

Figure 5 shows that the chemical bonding states of Co/C, Co/N-C, and Co/FN-C were analyzed using XPS. The C 1s XPS profiles of all samples exhibited three characteristic peaks, which corresponded to O=C-O/C-F groups (~288.95 eV), C-N groups (~286.47 eV), and C-C groups (~284.71 eV). Co/FN-C and Co/N-C showed larger C-N peaks compared to Co/C because of the additional N doping owing to the FNCQDs [43]. The N 1s XPS profiles showed the peaks of graphitic N (~400.80 eV), pyrrolic N (~399.99 eV), Co-N (~399.09 eV), and pyridinic N (~397.35 eV) (Figure 5b). Via the aromatic system, pyridinic N supplies two p-electrons, whereas pyrrolic-N provides one p-electron, thus resulting in increased catalytic activity [44,45]. Two core-level signal peaks at ~781.84 eV and ~797.50 eV were visible in the XPS spectrum of Co 2p (Figure 5d), corresponded to Co 2p_3/2_ and Co 2p_1/2_, and minor peaks of the Co^2+^ satellite (~802.67, ~786.86 eV), and Co-N (~779.31 eV). Furthermore, only Co/FN-C showed two peaks at ~684.6 and ~687.6eV in the F 1s XPS spectra, which represent the semi-ionic and covalent bonds of C-F, owing to the F atoms in the FNCQDs (Figure 5c). The doped F atoms can enhance the number of edge defects of the carbon lattice, thus providing more electrocatalytic active sites [46]. Furthermore, F anions provide redox-stable ligands, which can effectively improve catalytic stability [47,48].

Figure 6a–c shows the electrochemical activities of the Co/C, Co/N-C, and Co/FN-C electrodes measured via CV with different scanning rates from 10 to 100 m V s^−1^. All electrodes showed quasi-rectangular CV curves, which is characteristic of the charge and discharge process of electrostatic double-layer capacitors [49]. Moreover, we evaluated the electrochemical active surface area (ECSA), which is an indicator for comparing active sites of catalysts involved in the electrochemical reaction and is generally considered an important parameter for high ORR performance. In a without Faradaic potential window (0.1–0.3 V) [50,51,52,53,54,55], the C_dl_ (electrical double layer capacitance) and ECSA were tested and calculated by CV at different scanning rates as follows [56,57]:C_dl_ = j/r(1)
ECSA = C_dl_ /C_s,_(2)
where r corresponds to the scan rate, C_s_ corresponds to the specific capacitance (C_s_ = 0.035 mF/cm^2^), and j corresponds to the current density. Figure 6d shows that the C_dl_ values of the Co/C, Co/N-C, and Co/FN-C electrodes were 30.9, 45.4, and 62.9 mF cm^−2^, respectively [45]. Meanwhile, their ECSA values were 882.8, 1297.1, and 1797.1 cm ^2^, respectively (Figure 6e). The largest ECSA shown by the Co/FN-C electrode indicates that improved electrocatalytic active sites by the well-dispersed Co NPs on the carbon support increased the defects by N- and F-doping in carbon and effectively extended the electrochemically available active sites, unlike the cases with the other electrodes [58].

As shown in Figure 7a, to understand the electrochemical activity more comprehensively, LSV measurement was employed. The Co/FN-C electrode showed excellent ORR activities, including a limited current density of −4.678 mA/cm^2^, a half-wave potential (E_1/2_) of 0.702 V, and an onset potential (E_onset_) of 0.917 V, compared with the Co/C and Co/N-C electrodes (Co/C: a limited current density of −2.923 mA/cm^2^, E_1/2_: 0.702 V, E_onset_: 0.840 V; NCQD/Co/C: a limited current density of −4.678 mA/cm^2^, E_1/2_: 0.745 V, E_onset_: 0.888 V). Compared with other published papers, Co/FN-C sample outperformed the E_onset_ and E_1/2_ (Table 1) [50,59,60,61,62]. Furthermore, there are two main electronic pathways for the reduction of O_2_. In the direct 4-electron reaction, the single oxygen bond directly cleavage to generate H_2_O with high reaction efficiency, whereas the 2-electron reduction of the H_2_O_2_ intermediate is not only a low-efficiency ORR but also increases the ORR overpotential and corrodes the carbon catalyst support. Therefore, the reduction of H_2_O_2_ through the 4-electron pathway is conducive to the ORR [63]. The amount of oxygen reduction electron (n) was calculated through the Koutecky–Levich Formula (K–L) (3) and (4) [37].
(3)1J˙=1jd+1jk=1Bω12+1jk
(4)$$B=0.62^{n}FC_{0}{\left(D_{0}\right)}^{\frac{2}{3}V^{-\frac{1}{6}}}$$
where j is the measured current, j_d_ corresponds to the limiting current, j_k_ corresponds to the kinetic current, measured current, and limiting current, B corresponds to the proportionality coefficient, which is calculated using Equation (4); ω corresponds to the rate of angular rotation of the electrode (ω = 2 n*π/60, where n* corresponds to the number of rotations of the electrode per minute); n corresponds to the electrons transferred number when reducing one oxygen; C_0_ corresponds to the solubility of oxygen in the solution; v corresponds to the kinetic viscosity coefficient; Faraday constant is 96,485 C/mol; and D_0_ corresponds to the diffusion coefficient. To identify the ORR performance more clearly, the LSV result of the Co/FN-C electrode was measured at 100–1600 rpm (Figure 8a). According to the above formula, the electron transfer number in the 0.2–0.4 V (vs. RHE) range was 3.98 (about 4) (Figure 8b). Therefore, the Co/FN-C electrode induced an excellent ORR process in an acidic electrolyte owing to the 4-electron pathway, which was ascribed to the increased electrochemically active sites by the uniform dispersed of Co NPs on the carbon support and increased ORR activity by F and N doping in carbon [64,65]. Furthermore, to confirm the electrocatalytic stability, we compared the LSV curves of the three electrodes before and after CV for 5000 cycles (Figure 7b–d). After 5000 cycles, the Co/FN-C electrode showed the lowest potential degradation at ΔE_1/2_ = 2.1 mV compared with the Co/C (41.7 mV) and Co/N-C (5.4 mV) electrodes. The catalytic stability of the Co/FN-C electrode is excellent because F anions, as redox-stable ligands, are electron-withdrawing groups that effectively reduce the electron cloud density of carbon materials, thereby increasing their resistance to oxidative degradation. Furthermore, F anions can reduce the electron densities of metal nuclei ions and then increase the polarity of metal compounds, thereby improving the catalytic stability of the ORR. The following factors are responsible for this excellent electrochemical performance and stability: (1) the well-dispersed Co NPs, which increased the catalytic active sites; (2) F and N doping in carbon, which enhanced the catalytic activity; and (3) the redox-stable ligand by F anions, which reinforced the catalytic stability. To test the stability of the electrode in 0.1 M HClO_4_, we added Co/C, Co/N-C, and Co/FN-C to 0.1 M HClO_4_ overnight and examined their crystal structure by XRD. Appendix A shows the XRD patterns of before and after HClO_4_ treatment of Co/C, Co/N-C, and Co/FN-C samples. All samples showed a broad peak of diffraction at ~24.6°, which corresponds to the (002) plane of graphite (JCPDS 65-6212), and sharp peaks of diffraction at ~75.9°, ~51.3°, and ~44.2°, which corresponds to the (220), (200), and (111) planes of Co (JCPDS 15-0806), which means that the crystal structure of all samples was maintained after HClO_4_ treatment. Meanwhile, we add SEM-EDS mapping images and chemical composition before and after the electrochemical test of the Co/FN-C electrode, as shown in Appendix A. After the HClO_4_ treatment, the elements were well dispersed, and the atomic percentage of Co is maintained. Therefore, these results also indicate that the Co/FN-C sample can be electrochemically tested in HClO_4_.

## 4. Conclusions

In this study, we fabricated well-dispersed Co NPs on F and N carbon supports using ZIF-67 as a sacrificial template for an ORR catalyst. During the pyrolysis of ZIF-67 to obtain Co NPs with carbon support, the mixed FNCQDs alleviated the structural collapse of ZIF-67 and doped the carbon support with F and N atoms. The Co/FN-C electrode showed excellent ORR activities, including an ECSA of 3028.0 cm^−2^, a limited current density of −4.678 mA/cm^2^, E_1/2_ of 0.702 V, and an E_onset_ of 0.917 V. Simultaneously, the Co/FN-C electrode induced an excellent ORR in an acidic electrolyte because of the 4-electron pathway. Furthermore, after 5000 cycles, the Co/FN-C exhibited excellent long-term stability with an ∆E_1/2_ of only 2.1 mV. These performance improvements are attributed to: First, the well-dispersed Co NPs increase the active sites, thus alleviating the structural collapse of the CQDs during pyrolysis and improving the active sites number. Second, the F and N co-doping in the carbon support enhanced catalytic activity, which provided additional electrons and increased the number of edge defects. Finally, F anions, which served as redox-stable ligands, reinforced the catalytic stability. In conclusion, dispersing Co NPs uniformly in Co/FN-C is a prospective strategy for non-precious metal catalysts to be used in the ORR.

## Figures and Tables

**Figure 1 nanomaterials-13-02093-f001:**
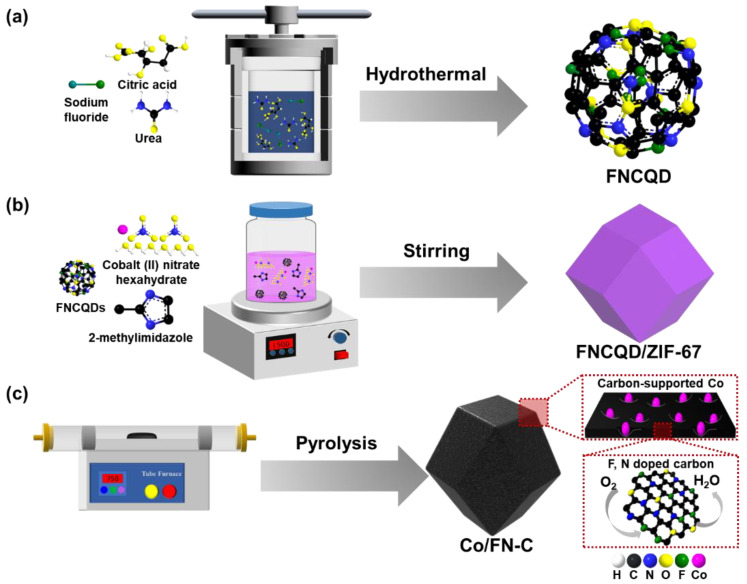
Schematic image of fabricating process about (**a**) FNCQDs, (**b**) FNCQD/ZIF-67, and (**c**) Co/FN-C.

**Figure 2 nanomaterials-13-02093-f002:**
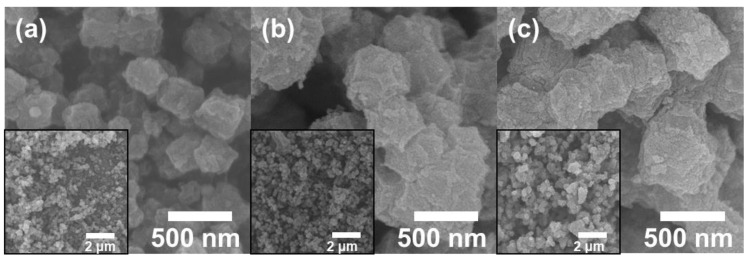
FESEM images for the as-prepared (**a**) Co/C, (**b**) Co/N-C, and (**c**) Co/FN-C.

**Figure 3 nanomaterials-13-02093-f003:**
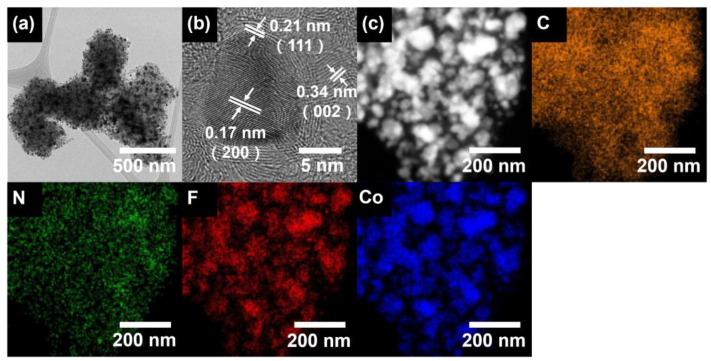
(**a**) Low- and (**b**) high-resolution TEM images, and (**c**) EDS mapping of C, N, F, and Co of Co/FN-C.

**Figure 4 nanomaterials-13-02093-f004:**
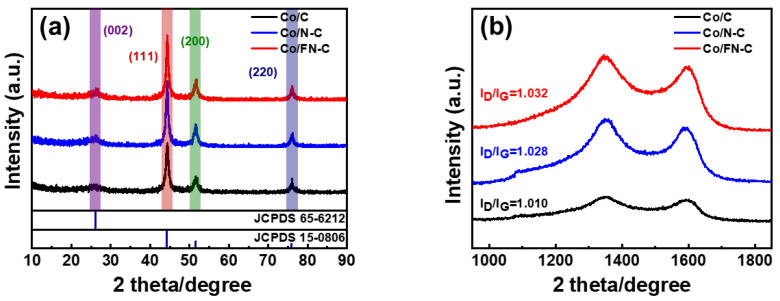
(**a**) X-ray patterns and (**b**) RAMAN spectrum of Co/C, Co/N-C, and Co/FN-C.

**Figure 5 nanomaterials-13-02093-f005:**
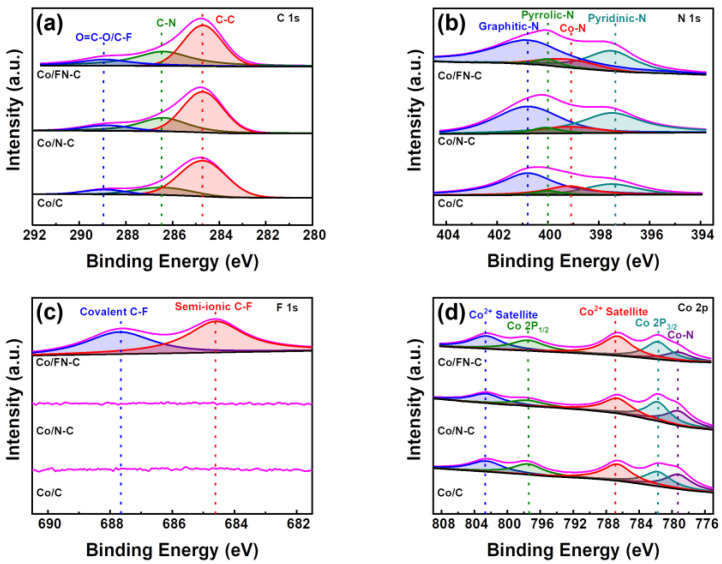
XPS spectrum (**a**) C 1s, (**b**) N 1s, (**c**) F 1s, and (**d**) Co 2p of Co/C, Co/N-C, and Co/FN-C.

**Figure 6 nanomaterials-13-02093-f006:**
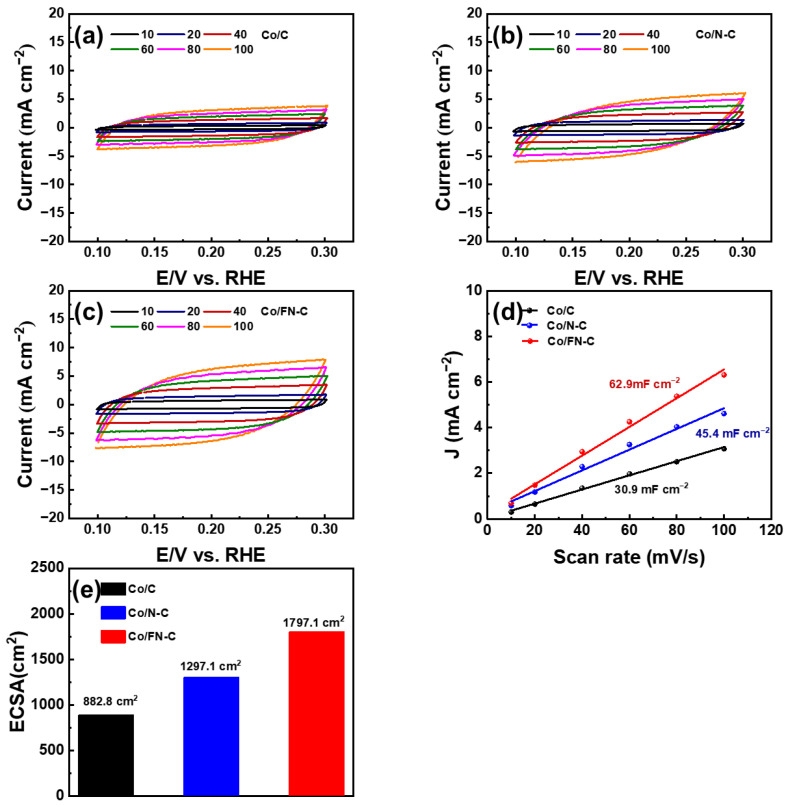
Cyclic voltammetry curve of (**a**) Co/C, (**b**) Co/N-C, (**c**) Co/FN-C (**d**) C_dl_ linear fitting (**e**) ECSA values for Co/C, Co/N-C, and Co/FN-C.

**Figure 7 nanomaterials-13-02093-f007:**
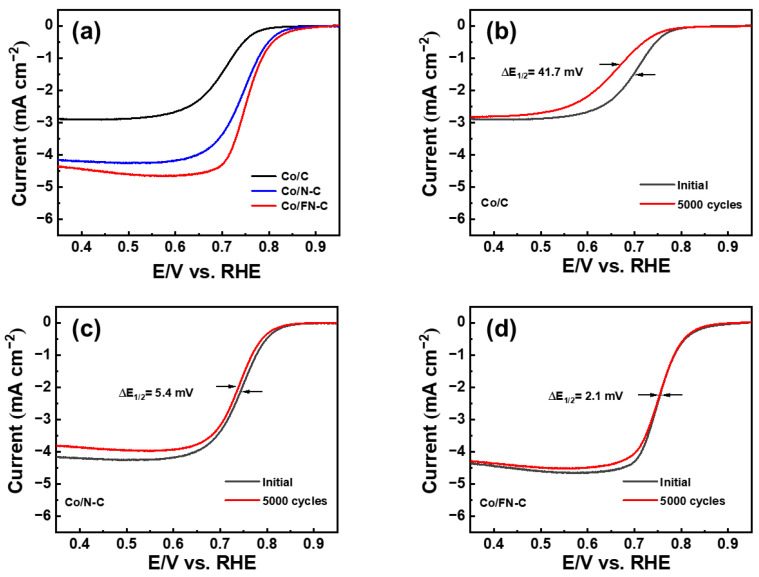
(**a**) Linear sweep voltammetry curves initial and ADT of (**b**) Co/C, (**c**) Co/N-C, and (**d**) Co/FN-C.

**Figure 8 nanomaterials-13-02093-f008:**
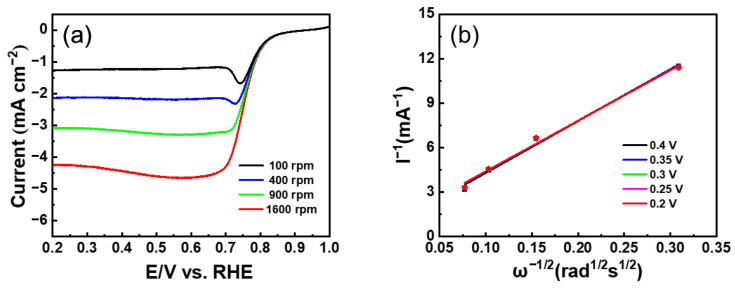
(**a**) LSV curves at various rotating speeds (**b**) Koutecky–Levich plots at different voltages of Co/FN-C.

**Table 1 nanomaterials-13-02093-t001:** Compared to other published papers the value of E_onset_ and E_1/2_.

Material	E_onset_ (V)	E_1/2_ (V)
Co-NSC 200	0.81	0.74
Fe-N-C	0.85	0.74
CAPANI-Fe-NaC	Not mentioned	0.73
PNGr	0.87	0.64
FeCo/C680	0.9	0.76
Co/FN-C	0.917	0.753

## Data Availability

All data generated or analyzed during this work are included in this published article.

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
