# Peer review of "MOF-Derived Co Nanoparticles Catalyst Assisted by F- and N-Doped Carbon Quantum Dots for Oxygen Reduction"

_nanomaterials, 2023, doi:10.3390/nano13142093_

Round 1

Reviewer 1 Report

The paper is very actual it was developed well-dispersed Co nanoparticles (NPs)  on the surface of a F and N co-doped carbon support (Co/FN-C) via the pyrolysis of ZIF-67 as a sacrificial template, while using F- and N-doped carbon quantum dots as a non-novel metal catalyst.

The paper can be published as presented.

Reviewer 2 Report

This paper reports synthesis of MOF-derived Co nanoparticles on F and N co-doped carbon support (Co/FN-C) and their catalytic performance for ORR. However, the papers contains obvious flaws that should be addressed or corrected before possible publication.

1. How did the FNCQD combine with ZIF-67? Were the FNCQDs absorbed on the surface of ZIF-67? Please provide more evidence for this since it is impotant to support their claims "The CQDs effectively maintained the structure of the sacrificial template of ZIF-67...".

2. Assignment of the XPS peaks in C1s and N1s should refer to some literatures, such as "J. Phys. Chem. C 2019, 123: 25570".   

3. ECSA should be obtained by scanning CVs within non-Faradaic potential range. For the Co/FN-C, ORR process occurs when the potential is under 0.9 V (vs. RHE). Therefore, the CVs within 0-1.2V cannot be used to determine Cdl and ECSA. 

4. Cobalt-based catalysts have been widely explored for electrocatalysis except ORR. Some recently published papers must be properly cited in the introduction section, such as "J. Colloid  Interf. Sci. 2021, 597: 361".

5. Some mistakes in grammar should be carefully checked and corrected, such as "Co/FN-C electrode an exhibited excellent onset" in line 243, P8.

6. ORR perfomance of the Co/FN-C catalysts should be compared with other published results.

Some minor mistakes should be carefully checked and corrected.

Reviewer 3 Report

        This work synthesized novel electrocatalysts for ORR by carbonizing the MOFs encapsulated with N-doped CQD or F-N-co-doped CQD, and found that the F-doped electrocatalyst can perform better. The idea is very interesting and novel. Experiments were systematically done. Based on the novel design of materials reported here, this work should be a nice work to be published. However, the design of electrochemical experiments has serious problems. Regarding the design of new materials, the material-characterization data are also not sufficient to fully support the conclusions. Several major issues need to be fixed; the synthesized materials should be fine, but the authors may need to redo all electrochemical experiments to fix these flaws. Thus, a major revision with 1-2 months of revision time, or a rejection and resubmission, is necessary before considering the publication of this work. Main issues are provided below.

1. All electrochemical measurements including ECSA and ORR tests were measured in 0.1 M HClO4 electrolyte. The pH of this electrolyte is close to 1, then how is it possible to keep the cobalt chemically stable? Cobalt-based compound is well known to show excellent ORR activity in alkaline solutions, but metallic cobalt, cobalt oxide, and cobalt hydroxide should be dissolved quickly once being immersed into the solution at pH 1. Thus, it is impossible to believe that all cobalt compounds are still present in the electrocatalyst during ORR. Some Co-N species may survive, but the majority of cobalt in the material should be dissolved in 0.1 M HClO4 even before all electrochemical tests. It is most likely that N-doped carbon or N-F-co-doped carbon is the real electrocatalyst for all electrochemical measurements here, with a fairly good activity and stability. The authors should carefully consider one of the following options:

(1) If the authors want to report the carbon-supported cobalt catalysts, all electrochemical experiments must be re-conducted in alkaline electrolyte.

(2) If the authors would like to keep the current ORR data, then the real electrocatalysts should be carbon-based materials with different dopants (and probably a minor amount of Co-N species). These 0.1 M HClO4-treated materials should be characterized and reported. The entire story of this manuscript should be rewritten as well.

2. Figure 6: There are obvious redox peaks in these CV curves, thus it is impossible to get ECSA from these CV data. The authors should collect CV data in the potential window without any Faradaic process in order to estimate the ECSA.

3. Characterizations of the materials before pyrolysis are all missing, which makes it impossible to know if the authors have successfully synthesized the CQDs, MOF, and CQD-encapsulated MOFs as claimed or not. The following experiments must be conducted and the data should be reported in the manuscript or SI:

(1) TEM images and elemental analysis (such as EDS) for FNCQDs and NCQDs.

(2) XRD data, nitrogen adsorption-desorption isotherms, and elemental analysis for ZIF-67, NCQD/ZIF-67, and FNCQD/ZIF-67. The porosity of the pristine ZIF-67 from the isotherm should be compared with the commonly reported value to verify the successful synthesis of the pristine MOF here.

4. The authors stated in the abstract that “the CQDs prevent structural collapse during the pyrolysis of ZIF-67”, which is obviously incorrect. All ZIF-67 crystalline structure has collapsed, as there is not any XRD peak of ZIF-67 in the XRD patterns of these pyrolyzed materials.

5. Minor point:

Line 74: “non-novel metal catalysts” It should be “non-noble”.

Round 2

Reviewer 2 Report

The authors have corrected their manuscript carefully and all of my concerns have been addressed properly. I can recommend it for publication in the current version.

Reviewer 3 Report

The authors have made significant efforts to address the previous comments with experimental data to support their claims. The work is solid and promising now. However, some data in the response to comments, i.e., the detailed material characterizations of the cobalt-based materials after treating in acidic solution for a long time, were not included in the revised manuscript and SI. The authors must report all data included in the response to reviewers with sufficient discussions into the revised manuscript or SI, before this paper can be accepted.

Author Response

We add the characterization of the cobalt-based materials after treating in acidic solution for a long time to manuscript and SI.
